# Association of the Dietary Inflammatory Index with Depressive Symptoms among Pre- and Post-Menopausal Women: Findings from the National Health and Nutrition Examination Survey (NHANES) 2005–2010

**DOI:** 10.3390/nu14091980

**Published:** 2022-05-09

**Authors:** Deniz Azarmanesh, Elizabeth R. Bertone-Johnson, Jessica Pearlman, Zhenhua Liu, Elena T. Carbone

**Affiliations:** 1Department of Human Nutrition and Hospitality Management, College of Human Environmental Sciences, University of Alabama, Tuscaloosa, AL 35401, USA; 2Department of Health Promotion and Policy, School of Public Health and Health Sciences, University of Massachusetts Amherst, Amherst, MA 01003, USA; ebertone@schoolph.umass.edu; 3Department of Biostatistics and Epidemiology, School of Public Health and Health Sciences, University of Massachusetts Amherst, Amherst, MA 01003, USA; 4Institute for Social Science Research, University of Massachusetts Amherst, Amherst, MA 01003, USA; jpearlman@issr.umass.edu; 5Department of Nutrition, School of Public Health and Health Sciences, University of Massachusetts Amherst, Amherst, MA 01003, USA; zliu@nutrition.umass.edu (Z.L.); ecarbone@nutrition.umass.edu (E.T.C.)

**Keywords:** dietary inflammatory index (DII), menopause, inflammation, depression, NHANES

## Abstract

During their lifetime, 20% of US women experience depression. Studies have indicated that a high Dietary Inflammatory Index (DII) score is associated with high C-reactive protein (CRP) levels and depression. No previous study has compared the association of the DII with different measures of depression (e.g., somatic, cognitive) among pre- and post-menopausal women. We used data from 2512 pre-menopausal and 2392 post-menopausal women from the National Health and Nutrition Examination Survey (NHANES) 2005–2010 database. We ran linear and logistic regression models to compare the association of the DII with survey-measured depression among pre- and post-menopausal women. We further assessed the mediation effect of CRP on the association of the DII and depression, using structural equation modeling. The odds of experiencing depression among pre-menopausal women was higher for all DII quartiles compared to the reference group (i.e., DII Q1), with an odds ratio (OR) of 3.2, 5.0, and 6.3 for Q2, Q3, and Q4, respectively (*p* < 0.05). Among post-menopausal women, only Q4 had 110% higher odds of experiencing depression compared to Q1 (*p* = 0.027). No mediation effect of CRP was found between DII and any of our depression outcome measures. Our findings suggest that lifestyle habits, such as diet, may have a stronger influence on mental health among pre-menopausal women than post-menopausal women.

## 1. Introduction

During their lifetime, 20% of US women experience depression. Women experience depression at a ratio of 2:1 compared to men [1], likely due to a higher chance of relationship distress, low social support, childhood adversity, obesity, physical inactivity [2], and sex hormone fluctuations during reproductive events in women (e.g., premenstrual, postpartum, menopausal transition) [3]. Menopause is associated with various changes, including an increase in inflammation [4], depression [5], and dietary habits [6]. Differences in depression levels among pre- and post-menopausal women may be due in part to differences in inflammation, with various studies finding higher levels of inflammatory biomarkers in post-menopausal women compared to pre-menopausal [7,8]. Estrogen may be protective against neurodegeneration by limiting the expression of neuroinflammatory mediators [9,10]. The decline in estrogen levels after menopause may contribute to the increase in inflammation and cognitive decline [11]. However, recent findings suggest that menopause hormone therapy may be effective in ameliorating depressive symptoms during peri-menopause, when hormone fluctuations are severe [12,13]. In fact, large prospective [14] and cross-sectional [15] research studies have shown that depressive symptoms are higher in hormone therapy users. Estrogen depletion also leads to central obesity, which exacerbates inflammation [11]. While hormone therapy has been ineffective in addressing depressive symptoms, an anti-inflammatory diet, rich in antioxidants, may be effective in opposing the inflammatory response to lack of estrogen, which may lead to depression in post-menopausal women.

Dietary components, such as saturated fats, omega-3 fatty acids, and fiber, can positively or negatively affect depression by inducing neuro-inflammation and stimulating changes in the hypothalamic-pituitary-adrenal (HPA) axis, neurotransmitters, and oxidative stress [16,17]. Pro-inflammatory diets, such as the Western diet, can raise the levels of pro-inflammatory biomarkers (e.g., C-reactive protein (CRP), interleukin 6 (IL-6)), which can increase the risk for the development of depression [17]. The Dietary Inflammatory Index (DII) assesses the effect of all pro- and anti-inflammatory dietary components in conjunction with each other [18]. The DII has been positively associated with inflammatory biomarkers such as CRP (i.e., a pro-inflammatory diet leads to a higher inflammation) [16,18,19,20,21,22,23,24]. Higher DII scores (i.e., more pro-inflammatory) are positively associated with various health conditions (e.g., depression) in women [24,25,26,27,28]. Diet may be one modifiable factor that contributes to higher inflammation and depression levels in post-menopausal women compared to pre-menopause. Data on the effect of diet on inflammation and depression by menopausal status are limited. A recent study found that the DII is associated with cognitive impairment in post-menopausal women [11]. No previous study has compared the association of the DII with inflammation and depression among pre- and post-menopausal women. 

We used the National Health and Nutrition Examination Survey (NHANES) to assess associations of the DII with inflammation and depression among pre- and post-menopausal women. We hypothesized that the association of the DII with inflammation and depression would be stronger among post-menopausal women as compared to pre-menopausal. We also tested the mediating role of inflammation, as assessed by CRP levels, on the association of the DII and depression in both population groups. 

## 2. Materials and Methods

We used three cycles of the NHANES database (2005–2010). We excluded participants below the age of 20, which is the cutoff recommended by NHANES for inclusion of adult participants. Individuals with a CRP value greater than 10 mg/L were excluded from analysis, as this is more likely to be an indicator of acute infection or inflammation [25]. Pregnant women were excluded, because pregnant women often make changes to their diet, and their inflammation and depression levels change during pregnancy [29,30,31,32]. Those with missing depression values, missing dietary intake or only one day of dietary intake reported, as well as missing covariates (i.e., body mass index (BMI), waist circumference, marital status, education, diabetes, race, smoking), were also excluded. There was a large number of missing values for poverty-to-income ratio (PIR) and physical activity (PA). To avoid further reducing our sample size, we included these as categorical variables in our regression models, with four categories based on the quartiles of the values and the fifth category including those with missing values. 

### 2.1. Assessment of Menopausal Status

To determine menopausal status, we used two questions from the NHANES Reproductive Health Questionnaire: (1) Have you had at least one menstrual period in the past 12 months? and (2) What is the reason that you have not had a period in the past 12 months? Participants were considered post-menopausal if the answer to the first question was “No” and the answer to the second question was “Menopause/Hysterectomy” (*n* = 2396). Participants were considered pre-menopausal if the answer to the first question was “Yes”, or if the answer to the first question was “No” and the answer to the second question was “Pregnancy”, “Breast feeding”, “Medical conditions/treatments”, and/or “Other”, and they were below the age of 55 years old (y) (*n* = 2512).

### 2.2. Assessment of Exposure: Diet

In NHANES 2005–2010, participants’ dietary intake was assessed using two previously validated 24 h dietary recalls (24DR) [33]. 

Inflammatory potential of the diet was assessed using the DII. The construct validity and calculation methods of the DII have previously been published [34]. The DII comprises 45 items, including macronutrients, micronutrients, and flavonoids. Using the 24DR of NHANES database, 27 of the 45 DII food items were available in our analysis to calculate the DII score. These food items were energy, carbohydrate, protein, total fat, saturated fat, cholesterol, omega-3, omega-6, monounsaturated fatty acid, polyunsaturated fatty acid, alcohol, caffeine, fiber, iron, magnesium, zinc, selenium, thiamin, riboflavin, niacin, folic acid, beta-carotene, and vitamins B12, B6, A, C, and E. We divided participants into quartiles of DII according to the distribution in our population. 

### 2.3. Assessment of Outcome: Inflammation

NHANES provides participants’ CRP levels as a measure of the body’s response to inflammation from inflammatory health conditions, such as arthritis. CRP was used as a continuous variable in our mediation analysis models.

### 2.4. Assessment of Outcome: Depression

NHANES used the 9-item Patient Health Questionnaire (PHQ-9) to assess depressive symptoms in the 2005–2010 cycles. The PHQ-9 is used for diagnoses of probable depression in primary care [35]. PHQ-9 has been previously validated and a score of greater than 10 has a sensitivity of 88% and a specificity of 88% for diagnosing moderate-to-severe depression [35]. 

PHQ-9 items and responses are coded on a four-point scale (Not at all = 0, Several days = 1, More than half the days = 2, Nearly every day = 3) [35]. The sum of all the item scores are calculated to make the PHQ-9 Score with the possible range of total scores of 0 and 27, respectively; the higher the scores, the more severe the symptoms. Participants had probable depression if they endorsed five or more of the PHQ-9 items for at least “more than half the days” in the past two weeks (i.e., total score > 10), and if one of the five symptoms was anhedonia or depressed mood [35,36]. If suicidality was present at all (i.e., score > 1), it was counted as one of the five items. We have used “probable depression” throughout this paper to refer to depression, as depression was determined using the PHQ-9 and not clinically diagnosed in this study. To calculate the somatic sub-score, we summed the PHQ-9 items on appetite changes, fatigue, sleep disturbance, and psychomotor delay/agitation [37]. We calculated the cognitive sub-score by summing the remainder of the five items on the PHQ-9 (i.e., anhedonia, concentration problems, depressed mood, low self-esteem, and suicidal ideation) [37]. The possible minimum and maximum of scores, respectively, were 0 to 12 for somatic depression, and 0 to 15 for cognitive depression.

### 2.5. Assessment of Covariates

In our regression models, we controlled for variables that were either associated with our dependent variable (i.e., depression) or were confounders in the association of the DII with depression. These variables included age, race/ethnicity, BMI, waist circumference (dichotomized into low risk and high risk based on the Centers for Disease Control and Prevention’s (CDC’s) cutoff of 35 inches for women), PA, smoking status, marital status, education level, and PIR. Mediation models that included CRP were further controlled for diabetes, as previous studies have shown insulin resistance affecting cytokine levels [27].

### 2.6. Statistical Analysis

We assessed the normality of distribution of the exposure and the outcome variable using histograms, and the linearity of association between covariates and our exposure and outcomes using scatter plots. We used pairwise correlations and variance inflation factors to test for possible multicollinearity between variables. To assess the distribution of subject characteristics, t-tests were used for continuous covariates across categories of DII and depression. We used the chi-squared test for our dichotomous outcome (i.e., probable depression) and one-way analysis of variance (ANOVA) to assess whether there were differences in values of the outcome variables and covariates across the DII quartiles.

We used logistic regression to test the association between the DII and the dichotomous outcome (i.e., probable depression). To test whether there was a significant difference in the odds ratio (OR) for probable depression between pre- and post-menopausal women, we ran a fully interactive model, where all the variables in our fully adjusted models were entered in the model in interaction with “menopausal status.” Linear regression models were used for our continuous outcomes: somatic and cognitive depression. We used structural equation modeling (SEM) to explore the mediating effect of inflammation on the association of the DII and our depression outcomes. By performing this analysis, we assessed how much of the association between the DII and depression was explained by the intervening effect of CRP. CRP and depression were entered in mediation analysis models as continuous variables, and the DII was entered as a categorical (i.e., quartiles) variable. 

We built two regression models: Model 1 was age adjusted; in Model 2, all covariates were included. 

All statistical analyses were conducted using the Stata Statistical Software (StataCorp. 2017. Stata Statistical Software: Release 16. StataCorp LLC., College Station, TX, USA). 

## 3. Results

Our final sample included 4908 pre- and post-menopausal women with mean age and standard deviation (SD) of 35.9 ± 9.6 and 62.9 ± 9.9, and mean (±SD) BMI of 27.7 (6.6) and 29.5 (6.5), respectively. The mean (±SD) waist circumference was 98.5 cm (±14.7) for both pre- and post-menopausal women. The mean (±SD) DII score was +0.1 (±1.8) and +0.007 (±1.8), the mean CRP was 2.5 (±2.4) and 2.8 (±2.3), and the mean PHQ-9 score was 3.6 (±4.4) and 3.5 (±4.5) for pre- and post-menopausal, respectively. The mean (±SD) somatic depression score was 2.1 (±2.4) for both pre- and post-menopausal women, and for cognitive score, the mean (±SD) was 1.4 (2.3) and 1.4 (2.4) in pre- and post-menopausal groups, respectively. Table 1 and Table 2 show subject characteristics by DII quartiles among pre- and post-menopausal groups, respectively. For pre-menopausal women, compared with Q1, those in Q4 of the DII were younger, less physically active, poorer, and had a higher BMI and waist circumference (WC). The same pattern existed for post-menopausal women; however, age was not significantly different across the quartile groups. 

Table 3 shows results assessing the association of the DII and probable depression. The mean PHQ-9 score increases across DII quartiles for both pre- and post-menopausal women. After adjustment for covariates (Model 2), there was a significant, linear association among both pre- and post-menopausal women (*p*-trend < 0.001). However, this association was significantly stronger in pre- vs. post-menopausal women (*p* interaction < 0.001).

Table 4 shows the results of our mediation analyses. CRP explained only 4% of the effect of change in the DII on depression symptoms score among pre-menopausal women (*p*-trend = 0.024) and 5% among post-menopausal women (*p*-trend = 0.006) in the age-adjusted model. There was no significant mediation effect of CRP in the fully adjusted model. Similar results were found for somatic and cognitive symptoms of depression.

## 4. Discussion

In this cross-sectional study, we found positive and significant associations between DII and the odds of having probable depression among pre-menopausal women, which was significantly different than post-menopausal women. The association of DII with somatic and cognitive depression was statistically significant but small in magnitude and unlikely to be clinically meaningful. Further, there were no differences between pre- and post-menopausal women in terms of these associations. The association of DII and depression was not explained by CRP.

Some studies have noted an increased risk for depression in menopausal transition (i.e., peri-menopause) and a decreased likelihood of depression after the onset of menopause [38]. Women in menopausal transition are two to four times more likely to experience a major depressive disorder and depressive symptoms [39]. In our study, we were not able to differentiate between pre- and peri-menopause, because that information was not available in the NHANES database. The stronger association that we noticed between DII and probable depression could be partially explained by the presence of peri-menopausal women in our pre-menopausal group. Future studies using the menopausal classification system set by Stages of Reproductive Aging Workshop (STRAW + 10) [40] may be helpful in better defining the menopausal status of women and minimizing misclassifications.

Menopause is concurrent with an increase in CRP [41] and markers of oxidative stress [42]. During menopause, the distribution of body fat changes and there is an increase in abdominal adiposity, which may result in higher inflammation [7]. Pro-inflammatory cytokines may affect mood and mental health through changes in the metabolism of neurotransmitters (e.g., serotonin, dopamine, noradrenaline, glutamate) [2,43,44]. However, we did not find CRP to be the mediating factor between the DII and depression. In fact, contrary to our hypothesis, we found the association of DII and probable depression to be stronger among pre-menopausal women than post-menopausal women. This suggests that depression may be more strongly related to lifestyle habits among pre-menopausal women and that depression after menopause is mainly associated with changes in hormonal balance. Cessation of estrogen production after menopause may be leading to neurodegeneration and depression [11]. It should be noted that the difference in mean CRP levels between the two groups in our analyses was small (i.e., 2.8 mg/L in post- and 2.5 mg/L in pre-menopause), which may have attenuated our results in detecting meaningful differences.

Future studies in different populations are needed to confirm our findings. The NHANES database only provides CRP as an inflammatory biomarker. CRP is viewed as a nonspecific biomarker for systemic inflammation [45] and a surrogate marker of cytokines, such as IL-6 and interleukin 1 beta (IL-1β) [46,47]. Even though CRP is an acceptable marker for the assessment of low-grade inflammation and may be used as a predictor for several chronic diseases [48], in the future, more studies should include other inflammatory biomarkers in their analyses, as measuring multiple inflammatory biomarkers is preferable in the assessment of inflammatory status [48].

It is important to note the limitations of our study. First, we did not have information on menopause hormone therapy among post-menopausal women. Not controlling for hormone therapy may have biased our results towards the null. Due to the formatting of medication use in the NHANES database, we were not able to exclude or control for contraceptives among pre-menopausal individuals. Sex steroid hormones can influence emotional processing in the cortical regions of the brain, and progesterone can lead to adverse mood effects among women [49]. We were also unable to exclude individuals using statins, which have been shown to have anti-inflammatory and antidepressive effects [50]. Use of these medications among our subjects may have confounded our results. Further, we were not able to control for anti-inflammatory medication use that may have had an impact on both inflammation and depression. Second, there is a possibility of residual confounding, such as social support, stressful life events, hormonal vulnerability [51], and history of depression, which can increase risk of depression among women, regardless of menopausal status [38,39,52]. Extreme hormonal fluctuations and experience of depression often occur during menopausal transition rather than late menopause [12,13]. The mean age of our post-menopausal group was 62.9 y, which is much higher than the average age of 49.9 y for the onset of menopause in the US [53]. Depression at an older age may be the result of chronic conditions, such as heart disease, which has higher incidence among women after menopause and has been associated with depression and anxiety among women [54]. We were not able to assess the correlation of DII, inflammation, and depression among peri-menopausal women, which is the stage of life when the effect of menopause on inflammation and depression may be most profound. Depression was not diagnosed clinically in our study, which may have led to misclassification of the population and outcome, biasing our results towards the null. However, the PHQ-9 used in the NHANES database has been validated, with high sensitivity and specificity for diagnosis of major depression, minimizing the possibility of measurement error in biasing our results. Due to the nature of our study design, we were only able to assess recent dietary intake, CRP, and depression levels. Depression and systemic inflammation are chronic conditions [25,26,27,55] and diet may affect these conditions over a long period of time. Lastly, due to the cross-sectional nature of our study, we were not able to infer causality. It is possible that the correlation of DII and depression may be explained by the association of depression with unhealthy diets. Previous epidemiological studies have suggested a bidirectional relationship between inflammation and depression [27], which further points to complicated physiological mechanisms in the association of these factors. Further, the wide confidence interval seen in the association of DII with depression among pre-menopausal women may be an indication that our models were underpowered to detect modest associations. 

There are also several strengths in our study. This was the first study to assess the association of DII and depression among pre-menopausal women, and the first to assess the mediation effect of inflammation, as assessed by CRP, on the association of DII and depression among both pre- and post-menopausal women. This was also the first study to compare these associations between the two groups to assess possible differences by menopausal status. Second, we used a large sample size, which provided a high statistical power and allowed us to consider various possible confounders in our models. Third, the population we used was a group of nationally representative women, making our results generalizable to American women.

Our results did not confirm that the association of the DII and depression is mediated by the body’s immune function associated with CRP. One possible explanation is that the DII may not be a robust dietary assessment tool in assessing the inflammatory potential of diets that correlate with the inflammatory biomarker levels. It is not clear what pathways mediate the association between the DII and depression. The DII may represent an overall healthy diet or be a proxy for healthy habits that are protective against depression (e.g., physical activity, social support). Diet may be related to depression via the neuroendocrine pathways affecting the HPA axis, neurotransmitters, and neurohormones. For example, excessive breakdown of tryptophan can result in surplus of metabolites, such as quinolinic acid (QUIN), which can lead to over-activity of the HPA axis, resulting in the development of depression [56,57]. More studies are needed to confirm these hypotheses.

## 5. Conclusions

The findings of our study do not suggest that the association of DII and depression is mediated by inflammation. We did not find any significant differences between pre- and post-menopausal women in terms of the associations of DII with various depressive outcomes. However, our results indicate that the inflammatory potential of diet may be more related to probable depression in pre-menopausal women than post-menopausal women. Future studies including other inflammatory biomarkers need to be conducted to confirm our findings. Future research on the effect of the inflammatory potential of diet on inflammation and depression among peri-menopausal women is warranted. Focusing on improving diet may be recommended as part of the treatment plan and prevention for depression before menopause.

## Figures and Tables

**Table 1 nutrients-14-01980-t001:** Subject characteristics by Dietary Inflammatory Index (DII) quartiles among 2512 pre-menopausal participants from the National Health and Nutrition Examination Survey (NHANES) 2005–2010.

		DII ^a^ Quartiles			
*n* (%)		Q1 670 (26.7)	Q2 634 (25.3)	Q3 621 (24.7)	Q4 586 (23.3)
Range of DII scores		−4.83, −1.21	−1.20, 0.18	0.19, 1.50	1.51, 4.49
**Subject Characteristics**		**Mean (SD)**	**Mean (SD)**	**Mean (SD)**	**Mean (SD)**
Age (y)		37.9 (9.5)	37.0 (9.6)	35.4 (9.4)	34.6 (9.7)
Physical Activity (PA) (MET-minutes) ^b^		942 (1392)	759 (1277)	579 (1179)	648 (1400)
Poverty-to-Income Ratio (PIR) ^c^		3.1 (1.7)	2.8 (1.6)	2.4 (1.6)	2.0 (1.5)
Dietary Inflammatory Index Scores		−2.1 (0.6)	−0.5 (0.4)	0.8 (0.3)	2.5 (0.7)
C-Reactive Protein		2.0 (2.1)	2.6 (2.4)	2.4 (2.3)	2.6 (2.4)
**Subject Characteristics**		** *n* ** **(%)**	** *n* ** **(%)**	** *n* ** **(%)**	** *n* ** **(%)**
Race	White	222 (55.3)	258 (44.8)	302 (42.6)	357 (43.2)
	Hispanic	111 (27.6)	189 (32.8)	240 (33.9)	235 (28.4)
	Black	42 (10.4)	97 (16.8)	125 (17.6)	194 (23.5)
	Multiracial	27 (6.7)	32 (5.6)	41 (5.9)	40 (4.8)
Smoking	Never Smoker	277 (68.9)	411 (71.3)	484 (68.4)	451 (54.6)
	Ever Smoker	125 (31.1)	165 (28.7)	224 (31.6)	375 (45.4)
Marital Status	Married/Partner	257 (64.0)	379 (65.8)	429 (60.6)	457 (55.3)
	Divorced/Widowed/Separated	52 (12.9)	81 (14.0)	96 (13.5)	119 (14.4)
	Single (Never Married)	93 (23.1)	116 (20.2)	183 (25.9)	250 (30.3)
BMI	Underweight and Normal Weight	200 (49.7)	220 (38.1)	280 (39.6)	307 (37.1)
	Overweight and Obese	202 (50.3)	356 (61.9)	428 (60.4)	519 (62.9)
Waist Circumference	Low Risk (≤35″)	225 (55.9)	253 (43.9)	320 (45.2)	358 (43.4)
	High Risk (>35″)	177 (44.1)	323 (56.1)	388 (54.8)	468 (56.6)
Education Level	Below High School	47 (11.7)	97 (16.9)	165 (23.3)	240 (29.1)
	High School Degree	56 (13.9)	90 (15.6)	153 (21.6)	214 (25.9)
	Some College/AA	122 (30.3)	205 (35.6)	233 (32.9)	272 (32.9)
	College Graduate and Higher	177 (44.1)	184 (31.9)	157 (22.2)	100 (12.1)
Quartiles of PIR ^d^	1	80 (19.9)	136 (23.6)	199 (28.1)	295 (35.7)
	2	68 (16.9)	115 (20.0)	163 (23.0)	208 (25.2)
	3	89 (22.1)	142 (24.6)	173 (24.4)	177 (21.4)
	4	141 (35.1)	144 (25.0)	130 (18.4)	98 (11.9)
	5 (Missing)	24 (6.0)	39 (6.8)	43 (6.1)	48 (5.8)
Quartiles of PA ^e^	1	94 (23.4)	190 (33.0)	275 (38.9)	311 (37.7)
	2	49 (12.2)	85 (14.8)	89 (12.6)	108 (13.1)
	3	100 (24.9)	121 (21.0)	157 (22.1)	163 (19.7)
	4	137 (34.1)	138 (23.9)	131 (18.5)	155 (18.8)
	5 (Missing)	22 (5.4)	42 (7.3)	56 (7.9)	89 (10.7)
Diabetes	Not Present	394 (98.0)	555 (96.4)	678 (95.7)	782 (94.6)
	Prediabetic/Diabetic	8 (2.0)	21 (3.6)	30 (4.3)	44 (5.4)

*p* values from analysis of variance (ANOVA) and chi-square tests were <0.05 for all comparisons. ^a^. Higher Dietary Inflammatory Index (DII) quartiles reflect higher inflammatory potential of the diet. ^b^. *n* for Physical Activity = 2303. ^c^. *n* for Poverty-to-Income Ratio = 2358. ^d^. PIR: Poverty-to-income ratio. Lower PIR quartiles reflect higher poverty status. ^e^. PA: Physical activity. Higher PA quartiles reflect higher PA levels. y, years old; MET, metabolic equivalent of task; BMI, body mass index; AA, associate degree; SD, standard deviation; ”, inches.

**Table 2 nutrients-14-01980-t002:** Subject characteristics by Dietary Inflammatory Index (DII) quartiles among 2392 post-menopausal participants from the National Health and Nutrition Examination Survey (NHANES) 2005–2010.

		DII ^a^ Quartiles			
*n* (%)		Q1 641 (26.8)	Q2 605 (25.2)	Q3 592 (24.7)	Q4 558 (23.3)
Range of DII scores		−4.49, −1.25	−1.24, 0.08	0.09, 1.47	1.48, 4.49
**Subject Characteristics**		**Mean (SD)**	**Mean (SD)**	**Mean (SD)**	**Mean (SD)**
Age (y)		64.0 (9.3)	62.5 (9.8)	62.8 (9.7)	62.5 (10.5)
Physical Activity (PA) (MET-minutes) ^b^		653 (1069)	457 (768)	384 (830)	270 (609)
Poverty-to-Income Ratio (PIR) ^c^		3.2 (1.5)	2.9 (1.5)	2.6 (1.5)	2.1 (1.4)
Dietary Inflammatory Index Scores		−2.2 (0.7)	−0.5 (0.4)	0.7 (0.4)	2.4 (0.7)
C-Reactive Protein		2.3 (2.0)	2.7 (2.2)	3.0 (2.3)	3.1 (2.4)
**Subject Characteristics**		** *n* ** **(%)**	** *n* ** **(%)**	** *n* ** **(%)**	** *n* ** **(%)**
Race	White	302 (69.4)	309 (57.8)	327 (49.7)	356 (46.5)
	Hispanic	65 (14.9)	133 (24.9)	170 (25.8)	218 (28.5)
	Black	49 (11.3)	78 (14.6)	146 (22.2)	173 (22.6)
	Multiracial	19 (4.4)	14 (2.6)	15 (2.3)	18 (2.4)
Smoking	Never Smoker	259 (59.5)	317 (59.3)	391 (59.4)	396 (51.7)
	Ever Smoker	176 (40.5)	217 (40.7)	267 (40.6)	369 (48.3)
Marital Status	Married/Partner	253 (58.1)	313 (58.6)	362 (55.0)	402 (52.5)
	Divorced/Widowed/Separated	153 (35.2)	194 (36.3)	256 (38.9)	318 (41.6)
	Single (Never Married)	29 (6.7)	27 (5.1)	40 (6.1)	45 (5.9)
BMI	Underweight and Normal Weight	153 (35.1)	143 (26.7)	146 (22.2)	183 (23.9)
	Overweight and Obese	282 (64.9)	391 (73.3)	512 (77.8)	582 (76.1)
Waist Circumference	Low Risk (≤35”)	147 (33.8)	122 (22.8)	140 (21.3)	171 (22.3)
	High Risk (>35”)	288 (66.2)	412 (77.2)	518 (78.7)	594 (77.7)
Education Level	Below High School	70 (16.1)	111 (20.8)	200 (30.4)	319 (41.7)
	High School Degree	98 (22.5)	143 (26.8)	170 (25.9)	214 (28.0)
	Some College/AA	123 (28.3)	163 (30.5)	191 (29.0)	169 (22.1)
	College Graduate and Higher	144 (33.1)	117 (21.9)	97 (14.7)	63 (8.2)
Quartiles of PIR ^d^	1	47 (10.8)	92 (17.2)	131 (19.9)	250 (32.7)
	2	98 (22.5)	118 (22.1)	176 (26.7)	214 (28.0)
	3	95 (22.9)	138 (25.8)	148 (22.5)	144 (18.8)
	4	154 (35.4)	152 (28.5)	142 (21.6)	102 (13.3)
	5 (Missing)	41 (9.4)	34 (6.4)	61 (9.3)	55 (7.2)
Quartiles of PA ^e^	1	141 (32.4)	212 (39.7)	319 (48.5)	392 (51.3)
	2	43 (9.9)	62 (11.6)	65 (9.9)	82 (10.7)
	3	122 (28.0)	115 (21.5)	116 (17.6)	125 (16.3)
	4	100 (23.0)	87 (16.3)	82 (12.4)	51 (6.7)
	5 (Missing)	29 (6.7)	58 (10.9)	76 (11.5)	115 (15.0)
Diabetes	Not Present	376 (86.4)	425 (79.6)	516 (78.4)	589 (77.0)
	Prediabetic/Diabetic	59 (13.6)	109 (20.4)	142 (21.6)	176 (23.0)

*p* values from analysis of variance (ANOVA) and chi-square tests were <0.01 for all comparisons, except for age (*p* = 0.07) and marital status (*p* = 0.26). ^a^. Higher Dietary Inflammatory Index (DII) quartiles reflect higher inflammatory potential of the diet. ^b^. *n* for Physical Activity = 2114. ^c^. *n* for Poverty-to-Income Ratio = 2201. ^d^. PIR: Poverty-to-income ratio. Lower PIR quartiles reflect higher poverty status. ^e^. PA: Physical activity; Higher PA quartiles reflect higher PA levels.

**Table 3 nutrients-14-01980-t003:** Estimated odds ratio for the association of Dietary Inflammatory Index quartiles and risk of probable depression, among pre- and post-menopausal participants from the National Health and Nutrition Examination Survey (NHANES) 2005–2010.

		Model 1 ^a^		Model 2 ^b^	
	Mean PHQ-9 Score (SD) ^c^	Odds Ratio	95% CI	Odds Ratio	95% CI
Pre-menopause (*n* = 2512)					
Quartile 1	2.7 (3.1)	1	–	1	–
Quartile 2	3.3 (3.9)	3.8	1.3, 11.3	3.2	1.1, 9.7
Quartile 3	3.6 (4.5)	6.6	2.4, 18.7	5.0	1.7, 14.3
Quartile 4	4.2 (5.0)	10.3	3.7, 28.4	6.3	2.2, 17.9
*p*-trend	–	<0.001		<0.001	
Post-menopause (*n* = 2392)					
Quartile 1	2.6 (3.2)	1	–	1	–
Quartile 2	3.2 (4.2)	1.8	0.8, 3.7	1.6	0.8, 3.5
Quartile 3	3.4 (4.3)	2.4	1.2, 4.8	1.7	0.8, 3.4
Quartile 4	4.4 (5.2)	3.7	1.9, 7.2	2.1	1.1, 4.3
*p*-trend	–	<0.001		0.026	

^a^. Model 1: Adjusted for age ^b^. Model 2: Adjusted for age, race (White, Hispanic, Black, Multiracial), BMI (underweight/normal weight, overweight/obese), waist circumference (≤35 inches (low risk), >35 inches (high risk)), marital status (Married/Partner, Divorced/Widowed/Separated, Never married), education (below high school, high school degree, some college/AA, college graduate and higher), smoking (never smoker, ever smoker), poverty-to-income ratio (quartiles), physical activity (quartiles) ^c^. Unadjusted. PHQ-9, 9-item Patient Health Questionnaire.

**Table 4 nutrients-14-01980-t004:** Mediation results examining CRP as a mediator of the relationship between Dietary Inflammatory Index (DII) quartiles and Depression Symptom Score using structural equation modeling (SEM), among participants from the National Health and Nutrition Examination Survey (NHANES) 2005–2010.

DII Quartiles	Total Effect Coefficient (SE)	95% CI	Direct Effect Coefficient (SE)	95% CI	Indirect Effect Coefficient (SE) (CRP)	95% CI	Indirect-to-Total Effect Ratio
Pre-menopause (*n* = 2512)							
Adjusted for Age							
1 (reference)	–	–	–	–	–	–	–
2	0.64 (0.22)	0.20, 1.09	0.58 (0.22)	0.13, 1.02	0.06 (0.02)	0.01, 0.12	0.09
3	0.99 (0.23)	0.54, 1.45	0.94 (0.23)	0.49, 1.40	0.05 (0.02)	0.006, 0.09	0.05
4	1.58 (0.24)	1.11, 2.06	1.51 (0.24)	1.04, 1.99	0.07 (0.02)	0.01, 0.12	0.04
*p*-trend	<0.001		<0.001		0.024		–
Fully Adjusted *							
1 (reference)	–	–	–	–	–	–	–
2	0.37 (0.22)	−0.06, 0.80	0.36 (0.22)	−0.7, 0.80	0.005 (0.01)	−0.02, 0.03	0.01
3	0.46 (0.22)	0.02, 0.91	0.46 (0.22)	0.02, 0.91	0.003 (0.007)	−0.01, 0.01	0.006
4	0.70 (0.24)	0.22, 1.18	0.69 (0.24)	0.21, 1.17	0.003 (0.009)	−0.1, 0.02	0.004
*p*-trend	0.010		0.10		0.684		–
Post-menopause (*n* = 2392)							
Adjusted for Age							
1 (reference)	–	–	–	–	–	–	–
2	0.48 (0.23)	0.02, 0.94	0.43 (0.23)	−0.02, 0.89	0.04 (0.02)	0.0003, 0.09	0.08
3	0.77 (0.22)	0.33, 1.21	0.68 (0.22)	0.24, 1.13	0.08 (0.03)	0.02, 0.15	0.10
4	1.70 (0.23)	1.24, 2.17	1.60 (0.23)	1.13, 2.07	0.10 (0.04)	0.03, 0.18	0.05
*p*-trend	<0.001		<0.001		0.006		–
Fully Adjusted *							
1 (reference)	–	–	–	–	–	–	–
2	0.15 (0.22)	−0.28, 0.59	0.14 (0.22)	−0.29, 0.58	0.004 (0.007)	−0.01, 0.02	0.02
3	0.23 (0.24)	−0.21, 0.66	0.21 (0.22)	−0.22, 0.66	0.01 (0.01)	−0.01, 0.04	0.04
4	0.86 (0.24)	0.39, 1.34	0.85 (0.24)	0.37, 1.32	0.01 (0.02)	−0.02, 0.05	0.01
*p*-trend	<0.001		<0.001		0.468		–

* Adjusted for age, sex, race (White, Hispanic, Black, Multiracial), BMI (underweight/normal weight, overweight/obese), waist circumference (≤35 inches (low risk), >35 inches (high risk)), marital status (Married/Partner, Divorced/Widowed/Separated, Never married), education (below high school, high school degree, some college/AA, college graduate and higher), smoking (never smoker, ever smoker), poverty-to-income ratio (quartiles), physical activity (quartiles), diabetes (non-diabetic, prediabetes/diabetes). For somatic depression, in the age-adjusted model, compared to Q1, the score was 0.76 units higher in Q4 among pre-menopausal women and 0.83 units among post-menopausal women (Appendix A). In the fully adjusted model, compared to Q1, the score was 0.31 units higher in Q4 (95% CI, 0.02, 0.61; *p*-trend = 0.050) among pre-menopausal women. Among post-menopausal women, the score was 0.37 units higher in Q4 compared to Q1 (95% CI, 0.08, 0.67; *p*-trend = 0.010). Compared to Q1, cognitive depression scores were significantly higher in Q4 for both pre- and post-menopausal women in the age-adjusted and fully adjusted models (Appendix A). Among pre-menopausal women, cognitive symptoms of depression were 0.42 units higher in Q4 vs. Q1 (95% CI, 0.13, 0.71; *p*-trend = 0.003) and was 0.32 units higher among post-menopausal women (95% CI 0.04, 0.61, *p*-trend = 0.016). SE, standard error; CI, confidence interval; CRP, C-Reactive Protein.

## Data Availability

Additional data can be found at Azarmanesh, D. The Association of the Inflammatory Potential of Diet with Inflammation and Depression among U.S. Adults: NHANES 2005–2010. Ph.D. Thesis. University of Massachusetts Amherst, Amherst, MA, USA, September 2020. https://doi.org/10.7275/18847643.

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
