# Peer review of "Association of the Dietary Inflammatory Index with Depressive Symptoms among Pre- and Post-Menopausal Women: Findings from the National Health and Nutrition Examination Survey (NHANES) 2005–2010"

_nutrients, 2022, doi:10.3390/nu14091980_

Round 1

Reviewer 1 Report

Summary of the study:

This is a retrospective cohort study using the NHANES 2005-2010 database to compare the association of the DII with survey-measured depression among pre- and post-menopausal women.

The odds of experiencing depression among pre-menopausal women was higher for all DII quartiles compared to the referent group.

There was no mediation effect of CRP found between DII and any of depression outcome measures. There was a significant modifying effect of DII and PHQ-9 by menopausal status.

Comments:

Thank you very much for conducting a well quality work using a large national database.

The analysis was well conducted using proper modeling and adjustment.

The manuscript was clearly written, and the authors concluded the study according to the results and addressed the study limitations appropriately.

Author Response

Dear reviewer,

On behalf of our team, I would like to thank you for your positive comments. We were delighted to learn that our manuscript was well received.

Dr. Azarmanesh

Reviewer 2 Report

This well-written paper reports on a secondary analysis of NHANES data (2005-2010) that examines the possible role of diet-sourced inflammation as measured by a self-reported index score (DII) in depressive symptoms in nearly 5, 000 female respondents categorized according to menopausal status (pre vs post).  Their hypothesis, that a stronger association would be observed between dietary inflammation and depressive symptoms in the postmenopausal group was not supported – with significant relationships being seen in all DII quartiles and depression only in the premenopausal group. Using CRP as a reasonable measure of inflammation, the investigators report no mediation effect. Despite these negative findings and the number of acknowledged limitations, the report is worthy of publication, as it points to ways to improve the designs of future menopause studies in this relatively new area of nutrition research. In other words, although it was reasonable to at least take a look, the NHANES questions on menopause and medication use are terribly imprecise and likely contributed to the negative findings for a menopause link. Moreover as the investigators note, it may be important to know how poorly the DII performed as a measure of inflammation (ie CRP). It is also notable that the results of this dissertation were presented at a peer-reviewed national meeting and published in abstract form.

There are a few concerns that need to be addressed to improve accuracy precision and clarity.

Intro: Although the premise is reasonable that pro-inflammatory diets may contribute to inflammation and depression (via obesity?) and that menopause enhances the inflammatory process, the blanket statement that previous studies have shown that menopause hormone therapy (the term ERT is no longer used) is protective against depression in postmenopausal women, is not accurate. Several large studies have produced conflicting data depending on the time since the last menstrual period, etc. Also ref #9 is not an appropriate reference for this statement

Methods: please state whether hormone use (contraceptives in the pre; MHT in the post) and statins wer included as an exclusion criterion…if not, this should be included as a significant limitation in the discussion

Results: No details, other than sample size, is provided for the full sample of 4908 respondents. It is always helpful to know the basic demographics (mean, SEs) of the population under study. Given the marked differences in age (approx. 30 yrs!), Im assuming there were a number of differences between the two groups, eg BMI, WC , as well as depression scores and CRP?, and these should be explicated if not in a table, at least in the text.

Tables 1 and 2: include in the legend the definition of quartile scores for DII, PIR and PA (higher is worse in all?). Also PA and PIR should be defined in the legend. Also, I understand when there are missing data in some of the variables the numbers may be less than the total ns in each category, but please explain how some of these ns (eg BMI, WC) are greater than the total Ns of each group??

Discussion:The explanation regarding the lack of mediation by CRP on the link between diet and depression is a bit unclear.  On the one hand you make the case that CRP increases after menopause (was this confirmed in your sample?)  but then you say that depression in postmenopause may be more associated with changes in hormone balance (ie falling estrogen elevates LDL, CRP, etc). Given that the mean values of CRP appear well below the threshold score of 10, is it possible that a relationship would be more difficult to demonstrate? The age difference is never really acknowledged as a limitation..ideally, your hypothesis that menopause influences the inflammatory impact on depression, needs to be tested in a younger group of postmenopausal women given the mean age of menopause is 52, and the age group here was approx. 62 when aging effects can swamp other influences.

Author Response

Dear Reviewer,

Thank you so much for your thoughtful comments. Please find our responses in the attached file.

Best,

Dr. Azarmanesh
